# Sign Language Recognition Method Based on Palm Definition Model and Multiple Classification

**DOI:** 10.3390/s22176621

**Published:** 2022-09-01

**Authors:** Nurzada Amangeldy, Saule Kudubayeva, Akmaral Kassymova, Ardak Karipzhanova, Bibigul Razakhova, Serikbay Kuralov

**Affiliations:** 1Faculty of Information Technologies, L.N. Gumilyov Eurasian National University, Nur-Sultan 010008, Kazakhstan; 2Institute of Economics, Information Technologies and Professional Education, Zangir Khan West Kazakhstan Agrarion-Technical University, Uralsk 090000, Kazakhstan; 3Department of Information and Technical Sciences, Faculty of Information Technologies and Economics, Kazakh Humanitarian Law Innovative University, East Kazakhstan Region, Semey 701400, Kazakhstan

**Keywords:** sign language, hand shape, palm definition model, MediaPipe Face, MediaPipe Hands, SVM, pattern, recognition, multiple classification

## Abstract

Technologies for pattern recognition are used in various fields. One of the most relevant and important directions is the use of pattern recognition technology, such as gesture recognition, in socially significant tasks, to develop automatic sign language interpretation systems in real time. More than 5% of the world’s population—about 430 million people, including 34 million children—are deaf-mute and not always able to use the services of a living sign language interpreter. Almost 80% of people with a disabling hearing loss live in low- and middle-income countries. The development of low-cost systems of automatic sign language interpretation, without the use of expensive sensors and unique cameras, would improve the lives of people with disabilities, contributing to their unhindered integration into society. To this end, in order to find an optimal solution to the problem, this article analyzes suitable methods of gesture recognition in the context of their use in automatic gesture recognition systems, to further determine the most optimal methods. From the analysis, an algorithm based on the palm definition model and linear models for recognizing the shapes of numbers and letters of the Kazakh sign language are proposed. The advantage of the proposed algorithm is that it fully recognizes 41 letters of the 42 in the Kazakh sign alphabet. Until this time, only Russian letters in the Kazakh alphabet have been recognized. In addition, a unified function has been integrated into our system to configure the frame depth map mode, which has improved recognition performance and can be used to create a multimodal database of video data of gesture words for the gesture recognition system.

## 1. Introduction

Globally, 432 million adults and 34 million children need rehabilitation for “disabling” hearing loss. It is estimated that by 2050, more than 700 million people—or 1 in 10 people—will have a disabling hearing loss. The prevalence of hearing loss increases with age, with over 25% of people over 60 years of age suffering from a disabling hearing loss [1].

Recently, more attention has been paid in the world to improving the quality of life of people with disabilities. Necessary conditions for movement, training, and interaction with the public for people with disabilities are being created; special hardware, software, scientific and technical products are being developed, and various social state programs and programs of inclusive education are being implemented. For scientists in countries around the world, creating a barrier-free society for people with disabilities is one of the most important tasks.

In modern Kazakhstan, one of the most important directions of state policy concerns the equal right to education for all citizens. The prerequisite for ensuring accessibility of education is an inclusive environment. The modern task of inclusive education provides intellectual development, ensuring equal access to education for all levels of the population, taking into account their psycho-physiological and individual characteristics. The process of inclusive education is conditioned by normative legal documents, such as the Law of the Republic of Kazakhstan on Education [2] and “the concept of development of inclusive education in the Republic of Kazakhstan” [3], which strengthen the requirements for professional activities of teachers, the process of barrier-free training of people with disabilities, and access to software products.

Object recognition technologies are a key factor that can provide solutions to improve the quality of life for people with disabilities. The ability of a machine to understand human gestures, interpret deaf-mute people’s intentions, and react accordingly is one of the most important aspects of human–machine interaction. At the same time, gesture recognition is quite a challenge, not only because of the variety of contexts, multiple interpretations, spatial and temporal variations, and complex non-rigid hand properties, but also because of different lighting levels and complex backgrounds. The first attempts to create various automated systems capable of perceiving the world like humans were made decades ago. Over time, greatly improved, these technologies have been widely used in many fields.

The authors proposed an algorithm for recognizing the dactylic alphabet of the Kazakh sign language. This algorithm can in turn can be used for recognition systems calculating Kazakh sign speech. The novelty of the proposed algorithm is that there are no full-fledged recognition systems of the Kazakh dactylic alphabet. Because of its similarity with the Russian sign language, the scientific community is limited to recognizing only the Russian dactylic alphabet, which consists of 33 letters, and the Kazakh dactylic alphabet of 42 letters, in which the letters i, ң, ғ, к, қ, ө, ë, ъ, and ь are denoted by several positions. The authors [4] compared Kazakh, Russian, English, and Turkish sign languages to prove that Kazakh Sign Language can exist as a separate sign language. A study of the form of the display was carried out in terms of configuration (arm/forearm), place of execution (localization), the direction of movement, nature of the movement, and a component that cannot be performed manually (facial expression and articulation). Despite the 50% similarity with Russian Sign Language, it can be said that Kazakh Sign Language is a separate language since, in turn, Russian Sign Language has 1050 words employed in the course of the study, about 30% of which are borrowed from English. The vocabulary of languages is many times less than the vocabulary of natural languages. In addition, during communication with each other, hard-of-hearing people continue to create new words, adapting them to the conversation. Therefore, we can conclude that the Kazakh sign language, in terms of vocabulary, is a separate sign language with its own specifics.

The next innovation of this work is that a unified “draw_people” functionality was integrated into the system for recording and demonstrating gestures in real life, in which users can set up a frame depth map mode, which in turn contributed to better results. The draw_people functionality makes it possible to obtain an approximately equal depth map of the dataset and the frame when shown in real time.

Moreover, linguists and sign language interpreters should also consider Kazakh Sign Language, as it is necessary to consolidate its status as a separate sign language and prevent its extinction.

## 2. Related Works

In order to find and determine the most appropriate and most optimal approach for the development of a full-fledged automatic sign language translation system, and for the development of appropriate digital devices based on gesture recognition methods to solve the previously identified social problem, work was carried out for the deaf and mute to analyze similar works based on machine learning methods (hereinafter ML).

It should be noted that there are a great number of types of tasks solved with ML, types of ML, and algorithms of ML models; therefore, a systematic special study of the application of ML for gesture recognition tasks is required. For a more complete characterization of the issue under consideration, the works of a number of authors have been studied. The scope of machine learning applications is very diverse. A number of scientific directions that are used in gesture recognition tasks can be distinguished: classical learning [5,6,7,8,9], ensembles [10,11,12], neural networks, and deep learning [13,14,15,16,17,18,19,20,21,22,23,24,25,26,27].

Classical learning: the simplest algorithms, characterized by direct heirs of computing machines of the 1950s. They knowingly solved formal problems, such as finding patterns in calculations and calculating the trajectory of objects. Today, methods based on classical learning are the most common. They form the recommendation module on many platforms.

When learning without a teacher, the machine itself must find the right solution among the cluttered data and sort objects by obscure features. For example, the machine may be required to classify a particular gesture among a set of data. The K-mean method is used in teacherless learning. Gani et al. [5] have applied the K-means algorithm in 2D space to divide all pixels into two groups corresponding to the hands of the signer. The K-means algorithm begins by placing K points (centroids) at random locations in 2D space. It places only two centroids that correspond to the user’s hands. Each pixel is assigned to the cluster with the closest centroid, and then the new centroids are calculated as the average of the pixels assigned to it. The algorithm continues until no pixel changes its affiliation to the cluster. If the distance between two centroids is less than a constant, they are combined into one.

When learning with a teacher, the machine has a instructor who knows which answer is correct. This means that the raw data is already marked (sorted) in the right way, and the machine only has to determine the object with the right attribute or calculate the result. The authors Sharma et al. [6] created a set of key descriptors with identical characteristics for each class of gesture image. This set is used to create a set of feature models for all training images. K-means clustering is performed to obtain K clusters with similar descriptors. Each image fragment is correlated with the closest cluster. Then for each image, all descriptors are compared to their nearest cluster, and a codeword histogram is created. A codeword dictionary is created using a set of feature histograms. In their approach, K is taken to be 150; i.e., 150 codewords are created for each image. Subsequently, different algorithms, including *k*-NN, were applied for classification. Arshad Malik et al. [7] proposed a system which captures the input data through a web camera without using any additional equipment, and then, using segmentation approach, the hand is separated from the background, and one can extract the necessary features from the image using principal component analysis (PCA). Finally, the gesture function is classified using K-nearest neighbors (*k*-NN). Anuradha Patil et al. [8] proposed a structure using Kinect with SVM, which is linear, and *k*-NN with weights. In general, their algorithm achieved moderate accuracy and speed in most conditions. Ramesh et al. [9] proposed an algorithm which consists of two steps: training and testing. In a training set of 50 different domains, video samples are collected. Each domain contains five samples, and each video sample is assigned a word class and stored in the database. The test sample is pre-processed using median filter, canny operator for edge detection, and HOG for feature extraction. The SVM receives the input data as HOG features and predicts the class label based on the trained SVM model. Finally, a textual description is generated in the Kannada language.

Works based on classical learning are limited in the number of recognized gestures, mostly applied to dactyl alphabets of gesture languages; that is, limited in the amount of tested data, in real-time work, to a small number of gestures. Many works used different algorithms for feature extraction and classification, and they are also often used for recognition of static gestures.

Ensembles: groups of methods that use several machine learning methods at once and correct each other’s errors. They include such classifiers as Random Forest and XGBoost, which boost when algorithms are trained sequentially, with each one paying special attention to the errors of the previous one.

Qin et al. [10] proposed a method of gesture recognition based on the fusion of several spatial features. These spatial features describe the shape and distribution of gestures in the local space, and one performs feature filtering, preserving the features of the discriminant information to reduce the computational cost. We have experimented with two large sets of gesture data and, compared to popular methods, our method effectively improves recognition quality. In the future, we will consider how to improve the features to make them more intelligible; for example, by using convolutional neural networking and other methods to automatically learn gesture characteristics. Su et al. [11] proposed a random forest-based Chinese Sign Language (CSL) sub-word recognition method using an improved decision tree to increase the probability of obtaining the correct result from each decision tree in random forests. Based on the recognition results of 121 frequently used CSL sub-words, the superior performance of the random forest method in terms of accuracy and reliability was tested. Results with a recognition accuracy of 98.25% were obtained. Su et al. [11] proposed a system in which they introduced a two-stage pipeline based on two-dimensional body connection positions extracted from RGB camera data. First, the system divides the signed expression data stream into meaningful word segments based on a frame-by-frame binary random forest. Each segment is then converted into an image-like form and classified using a convolutional neural network. The proposed system is then evaluated on a data set of continuous Japanese gesture language sentence expressions with variations of non-manual expressions. By exploring a variety of data representations and network parameters, we can distinguish verbal segments of specific non-manual intonations from the underlying body joint motion data with 86% accuracy.

Kenshimov et al. [12] proposed a system of dactylic alphabet recognition of Kazakh Sign Language based on SVM, Extreme Gradient Boosting, and Random Forest. The Kazakh Sign Language dactyl alphabet has 42 letters, but in their work 31 classes were distinguished; that is, two-handed and dimaic gestures were not included, in contrast to our system.

Ensemble methods are a machine learning paradigm in which multiple models (often called weak learners or baseline models) are trained to solve the same problem and combined to improve performance. The basic hypothesis is that if we combine weak learners correctly, we can obtain more accurate and/or reliable models.

Neural networks and deep learning: the most complex level of AI learning. Neural networks simulate the work of the human brain, which consists of neurons constantly forming new connections with each other. They can be conventionally defined as a network with many inputs and one output. Neurons form layers through which a signal sequentially passes. All this is connected by neuronal connections, or channels, through which data are transmitted. Each channel has its own “weight”—a parameter that affects the data it transmits.

The AI collects data from all inputs, evaluating their weight according to given parameters, and then performs the desired action and outputs the result. At first, it is random, but then, through many cycles, it becomes more and more accurate. A well-trained neural network works like a normal algorithm, or more accurately.

The real breakthrough in this field has been deep learning, which trains neural networks at multiple levels of abstraction.

Deep neural networks are the first to learn how to recognize gestures, one of the most complex objects for AI. They do this by breaking them into blocks, identifying the dominant lines in each, and comparing them to other images of the desired object [12,13,14,15,16,17,18,19,20,21,22,23,24,25,26,27].

Recurrent neural networks [16,28] are mainly used for text and speech recognition. They identify sequences in them and associate each unit—a letter or sound—with the rest.

For machine learning algorithms, it is important that the data arriving to the input of the algorithm can accurately describe the properties of the object, provide accurate information about the object, and the volume of incoming input data. Because the amount of incoming data depends on the speed of data processing, the requirement for machine performance and the accuracy of object recognition depends directly on the accuracy of the input data. The MediaPipe technology used in this work allows one to solve these problems, and at the input of the artificial neural network it provides only the coordinates of 21 points and the trajectory of change of each point. In addition, it removes the load on the algorithm used.

In their study, Nafis and Ayas Faikar used the wrist position recommended by MediaPipe. The Shift-GCN model included modification of the moving weight of the main points of the obtained palmar joints. The study used the values of the main points of the hand as a data set [18]. Caputo and Ariel created the SHREC 2021: Track system, which recognizes hand gestures based on the hand skeleton. With Leap Motion, they created many datasets for 18 character classes, and the datasets were learned and recognized by the ST-GCN model. The authors wrote that there were some errors in the recognition of dynamic gestures [29].

Halder et al. proposed a method based on the open-source MediaPipe platform and a machine learning algorithm. The model is lightweight and can be adapted to a smart device with American, Indian, and Turkish sign languages serving as the data set. The reliability and accuracy of the proposed models are estimated at 99% [30]. The algorithm proposed by Gomase and Ketan using Mediapipe and recognition using computer vision was partially successful, and accurate at an average of 17 frames per second, with an average accuracy of 86 to 91% [31]. In their papers, Alvin and Arsheldy proposed an American Sign Language recognition system based on Mediapipe and K-mean. Thus, Mediapipe is one of the most advanced real-time gesture recognition technologies [32].

Chakraborty proposed a methodology for classifying English alphabets rendered by various Indian Sign Language (ISL) hand gestures using the Mediapipe Hands API launched by Google. The purpose of using this API is to find the 21 significant points in each hand along with their x, y, and z coordinates in 3D space. Due to the lack of a proper dataset available on the internet for ISL, at the very beginning, they created a dataset of 15,000 per English character, each consisting of the coordinates of 21 points recognized by the Mediapipe Hands API [33].

The listed studies [29,30,31,32,33] have significantly contributed to the development of multimodal gesture corpora. However, the problem of calculating the frame depth map for gesture recognition using the Mediapipe technology is still relevant. The image data is first acquired with a simple camera. The coordinates of the human palm joints are then computed using the BlazePalm single finger detector model, which is available in MediaPipe. In Kazakh Sign Language, SVM was used for multiple classifications of various numbers and letters. The novelty of the proposed algorithm is that, firstly, there are no full-fledged systems for recognizing the Kazakh dactyl alphabet today, and secondly, a unified draw_people functionality was integrated into the system for recording and demonstrating gestures in real time. Moreover, users can configure the frame depth map mode in that mode, contributing to the achievement of superior results.

The task of creating universal multimodal gesture corpora arises due to the solution of several unimodal subtasks: recognizing hand gestures, identifying the movement of the body and head, and recognizing facial emotions. The listed tasks are fraught with problems of spatial and temporal variations and complex non-rigid properties of hands, different levels of illumination, and complex backgrounds. Recognizing hand gestures is a semiotic task in which the dactyl alphabet and tracing speech are used. The dactyl alphabet is used for the introduction and transmission of the sound of new words (for example, proper names), for which there are no ready-made means of sign language. Tracing signed speech is a secondary sign system that traces the sounding language’s linguistic fabric.

S. Zhang et al. [34] propose a sign language recognition structure that combines RGB-B input and two-stream space–time networks. The ARS approach covers key information for aligned multimodal input data and effectively eliminates redundancy. The local focus of the hand optimizes the input of the spatial network. In addition, the D-shift network generates depth movement features to investigate depth information efficiently. Subsequently, convolution fusion is performed to merge the two feature streams and improve recognition results. Yu et al. [35] proposed the SKEPRID system, a repeated recognition method resistant to significant body posture and lighting changes. By including information about the skeleton, they reduced the influence of various poses and developed a set of light-independent features based on the skeleton, significantly increasing the accuracy of repeated recognition.

Luqman et al. [36] presented a new multimodal video database for sign language recognition. Unlike existing databases, the database focuses on signs that require both manual and non-manual articulators, which can be used in various studies related to sign language recognition. Two cases were considered for sign-dependent and sign-independent modes using manual and non-manual signs. In the first case, we used color and depth images directly, while in the second, we used optical flow to extract more relevant features related to the signs themselves and not to the signatories. The best results were obtained using MobileNet-LSTM with transfer training and fine-tuning: 99.7% and 72.4% for the “sign-dependent” and “sign-independent” modes, respectively. Kagirov et al. [37] also presented the Russian multimedia database of the Russian sign language. The database includes lexical units (individual words and phrases) from the Russian sign language within one thematic area, called “food in the supermarket”, and was collected using the MS Kinect 2.0 device with Full HD video modes and depth maps. They provide new opportunities for a lexicographic description of the vocabulary of Russian sign language and expand research in the field of automatic gesture recognition.

D. Ryumin et al. [38] proposed an approach for the detection and recognition of 3D gestures of one hand for human–machine interaction. The logical structure of the system modules for recording the gesture database is described. The logical structure of the 3D gestures database is presented. Examples of frames demonstrating gestures in full high definition format, in map depth mode, and the infrared range are given. Models of a deep convolution network for recognizing faces and hand shapes are described. The results of automatic detection of the area with the face and the shape of the hand are given.

The works 44–48 listed above aim to calculate depth using a D-shift network that generates depth movement features, or a Kinect camera that provides depth information to increase gesture recognition. However, in our work, we calculate the depth of the frame using a simple draw_people functionality, which also contributed to the improvement of the indicator without using additional models or special cameras.

## 3. Problem Description and Proposed Solution

### 3.1. Problem Description

Gesture (lat. Gestus—body movement)—movement of the human body or its parts, which has a certain meaning, i.e., is a symbol or emblem.

Sign language is a method of interpersonal communication, characterized by specific lexical and grammatical patterns, supported by gestures of hearing-impaired people.

Sign language is a system of non-verbal communication between people with normal hearing and people with hearing impairments, and the latter is actually used as the main mode of communication, in which one can find a gesture that corresponds to each word [15]. The basic unit of sign language is a gesture, i.e., the ability to indicate an object through gestures, facial expressions and articulation, head turning, etc., visualization of object parameters.

In most cases it is not possible to convey names, foreign, technical and medical references with the help of sign language. Therefore, along with sign language, the deaf (hearing-impaired) widely use dactylic alphabet as a supplement (Figure 1).

The grammar of the dactylic language resembles that the grammar of the native language of the deaf. Dactylogy can often be described as writing with fingers in the air: visual perception and reliance on all the rules of spelling, such as writing. But not punctuation marks: exclamation and question marks are conveyed through appropriate facial expressions; period and multi-point pauses; dashes, colons and other punctuation marks, although peculiar types of expression, are not indicated in dactyl writing.

To parameterize gesture demonstration, five components of gesture are distinguished: configuration (hand/forearm), place of performance (localization), direction of movement, nature of movement, and a component not performed by hands (facial expression and articulation) [39,40]. Let’s take localization, movement and direction of the palm as the basic properties in the gesture demonstration, and introduce the following concept and notations for the model construction (Table 1).

Based on the features entered in Table 1, we can determine the complexity of the input data to solve the problem of sign language recognition. On the basis of the parameters designated in this table, the classification of words of the Kazakh sign language is carried out. For the sign language recognition task, the place of the gesture (Figure 2) is very important, because if the gesture is performed in a neutral area, you can separate the object from the background by putting on the speaker clothes of the same color, but if the place of display is marked by touching the face, neck, it is difficult to separate the hand from the face or neck. If the display location is at the waist or around the shoulders, you must have certain requirements for the background in order to separate the object from the background.

General gestures can be divided into static and dynamic gestures. Static gestures represent the position of the hand without any movement in space, and dynamic gestures are characterized by sequential hand movements from the starting point to the end point over a certain period of time (Figure 3). At the same time, many letters of the dactyl alphabet can be referred to static signs (Figure 4).

When demonstrating the gestures «eкi(two)», «қaзaн (caldron)», «қыcқaшa (briefly)», «ipi (large)» in gesture language, the hand takes only one position, and when demonstrating dynamic gestures, not only changes the hand position but also the configuration, as well as all the changes from Table 1.

Gestures with two hands are called symmetrical if the shape and direction of movement of two wrists coincide or reflect each other’s movements. In the words «дәлдiк (accuracy)», «бapaбaн (drum)» depicted in the pictures, both hands move symmetrically to each other. In asymmetric two-handed gestures, one hand often does not move, or moves in a different direction—this hand is called the passive hand, and the other hand can perform complex movements—this hand is called the active hand, often (it allows you) to determine the shape and movement of the active hand.

For static and dynamical gestures with the palm pointing to the camera or to the speaker (Figure 4 and Figure 5), various algorithms can be used to accurately determine the configurations or shape of the hand [41].

If the palm is oriented left, right, up and down, the hand configuration may not be correctly read by a simple camera and thus impairs the detection of hand features and object recognition (Figure 6).

In this paper, we proposed an algorithm for recognizing one-handed and two-handed gesture types that satisfies the following conditions:Ω HA ψ PTFS↔ ML/DULRSΩ HA/OH ψ PTFS↔ ML/DULRSΩ HA/RLH ψ PTFS↔ ML/DULRSΩ HA/TF ψ PTFS↔ ML/DULRSΩ HA/TN ψ PTFS↔ ML/DULRSΩ NZ ψ PTFS↔ ML/DULRSΩ NRLSH ψ PTFS↔ ML/DULRSΩ W ψ PTFS↔ ML/DULRS

### 3.2. Proposed Solution

For a gesture recognition system, the technologies that are used to collect raw data on hand movements, facial expressions or body language play a crucial role. In general, input data acquisition devices for gesture recognition systems fall into two categories: simple cameras and various sensors. Accordingly, it can be said that the methods and algorithms used in gesture recognition are directly dependent on these data collection devices.

The MediaPipe technology used in this paper, without special sensors and gloves, using a simple camera, can ensure information about the main points, characteristics and the position of the hand, which can be provided by sensors and devices, using a simple camera.

Since the 1990s, the use of special gloves in gesture recognition applications has been widespread, and there has been great interest in methods using various sensors. The solution to the problem of gesture recognition using sensors is still relevant and has been described in many modern works.

Saggio et al. [42] proposed a system based on wearable electronic devices and two different classification algorithms. The system has been tested on 10 Italian sign language words: “*costo*”, “*grazie*”, “*maestro*”, as well as on international words such as “Google”, “internet”, “jogging”, “pizza”, “TV”, “Twitter”, and “*ciao*”. Hou et al. [43] proposed a SignSpeaker system based on a smartwatch. Hou described how each sign has its own specific motion model and can be converted into unique gyroscope and accelerometer signals. They implemented their system on a pair of ready-made commercial devices—a smartwatch and a smartphone.

The FinGTrAC system [44] demonstrates the feasibility of fine-grained finger gesture tracking using a minimally invasive wearable sensor platform (a smart ring worn on the index finger and a smartwatch worn on the wrist). The main contribution is to increase the scale of gesture recognition to hundreds of gestures using only a rare set of wearable sensors. In contrast, previous work detected only dozens of hand gestures. Such rare sensors are comfortable to wear, but they cannot track all fingers and provide insufficient information.

Yin et al. [45] proposed a gesture recognition system based on an information glove; their proposed glove included a FLEX2.2 sensor and an STM32 chip to detect finger position. The data received from the sensor is fed to the input of the ANN and a signal is recognized as a result of comparison with a reference. Similarly, Bairagi [46] et al. used an information glove and mounted an ADXL335 accelerometer sensor and a resistance sensor. The data from the accelerometer was converted into digital data using an ADC and recognized using a microcontroller. It was then sent to the Android device via a Bluetooth module and converted from text to speech.

Chiu [47] et al. proposed a gesture recognition system using an autonomous current source attached to the back of the hand; that is, the gesture was recognized by detecting the movement of the joints of the hand using a triboelectric nanogenerator.

Mummadi et al. [48] have developed a data processing glove design that detects fine-grained hand shapes on gloves, based on IMU sensors on all fingertips. It takes advantage of the latest System-on-Chip designs that are powerful enough to perform real-time data fusion and classification procedures in the glove. A total of five IMUs, a multiplexer, and an embedded microprocessor make up the entire configuration of the glove. To improve the noisy and drift-prone sensor readings from the IMUs, an additional filter was used to generate a smooth and consistent signal. In addition, combining the data allows accurate measurement of orientation and finger movements.

The aforementioned research on glove use has made a major contribution to gesture recognition. However, the problem of the number of recognized gestures is still relevant, since there is a limited amount of test data, despite the fact that many of them are effective and show results above 90%. In addition, the systems in question use gloves and additional devices to obtain information about hand joints and finger positions, and these devices are known to be very expensive and inconvenient to use in the household. The MediaPipe technology [49] used in our work is a revolutionary product in the field of gesture recognition because it does not require the use of additional, very expensive, or unavailable devices to formalize and track palm and finger movements.

Many current automatic sign language translation systems, especially those based on machine learning, require the performance of processing; therefore, the architecture and levels of the artificial neural network are large enough to recognize dynamic gestures.

The algorithm we proposed does not require processing performance and can also be updated for mobile gadgets (Figure 7). This algorithm, based on the MediaPipe Hands technology and the OpenCV library, was able to recognize two-handed and one-handed gestures in real time and provide reliable results. MediaPipe Hands is one of the best solutions for hand and finger recognition.

#### 3.2.1. Get Image

Within the framework of the article, a gesture recording functionality was developed, in which the active region (ROI) and depth map mode can be configured, which in turn significantly improved the results of gesture recognition compared to previous similar systems [29,30,31,32,33]. This functionality has been integrated into the gesture recording module and into the real-time dactyl letter recognition module.

The gesture recording system consists of the following steps:(1)Starting the camera(2)Clicking on the screen(3)The layout of the human upper body appears (Figure 8)(4)Adjusting the layout(5)Starting recording

As emphasized in the introduction, it was essential for us to develop budget systems with an ordinary everyday camera without additional devices. The recording was made by a regular Logitech WebCam C270 USB webcam, which allows one to receive 1280 × 720 video with a frequency of up to 30 frames. The system works with any camera above 480 × 640.The proposed system works in real time, so it is necessary to record, analyze, and pre-process video in a short time and feed the information to the input of the model. In the Get Image module, recording is performed in real time until 500 frames are reached, or until the speaker presses the letter q. Every 5th frame out of 500 recorded frames is saved. Thus, 100 frames are recorded for each 500 frames. A total of seven people (four of 20 years of age, one of 40 years of age, one of 11 years of age, and one of 8 years of age) participated in the experiment, and 4100 frames of the KSL alphabet and 1500 frames of numbers from 1 to 15 were captured for each.

#### 3.2.2. Get Data

Recognizing human hand configuration and direction of movement is one of the vital components, as it opens up possibilities for natural human–computer interaction. Among the five components described above, the configuration, direction of movement, and character of movement are important components, since the main information about the gesture is read from them. Correspondingly, reading the coordinates of the finger joints and reading the trajectory of each finger can be a prerequisite for calculating gesture language recognition.

To determine the initial location of the hand, we use the BlazePalm single-finger detector model which is available in MediaPipe. Hand detection is quite a complex task: the system must process hands of different sizes, in different lighting, on different backgrounds, with crossed fingers or closed hands, etc. In addition, after separating the palm in the frame from the frame, the system processes only the values inside the frame, which, in turn, reduces the load on the performance of the machine by a multiple of five to six.

The Get Data module reads a static frame, then converts the image from BGR to RGB, outputs a *Y*-axis image, sends it to the MediaPipe input, and detects a palm.

The defined palm is a list of 21 base coordinates, in which each point consists of x, y, and z values; x and y are set to the width and height of the image, respectively [0, 0, 1.0], and z to the depth of the point. The forearm depth is the reference point, and the smaller the value, the closer the orientation is to the camera. The z value uses approximately the same scale as x.

After obtaining the coordinates of 21 joints of the human palm, these values are transferred to the real coordinates of the virtual three-dimensional world, and the hand orientation is determined based on magnetic positioning (Figure 1). There is a terminology for the digital three-dimensional space coordinate system. This is unconventional, although these concepts help programmers create 3D applications and games. Here, x is width, y is height, and z is depth compared to mapping.

From the 100 frames recorded in the Get Image module, the palm is detected by BlazePalm, and the palm is extracted from the frames with the detected palm (Figure 9). A csv data file will be created as a result of palm detection frames.

When submitting input data into an artificial neural network, it is important to process the input data correctly, not the neural network itself. Accordingly, it is first necessary to analyze the correlation and select the correct data. To realize this goal, data visualization is performed (Figure 10). Knowing how to choose the right type of graph is a key skill, because distortion can lead to misinterpretation of the data obtained by qualitative data analysis. Therefore, the data were visualized using two tools.

For example, if class “4” initially has 100 frames, palms are detected from about 70 of these frames, and hand values are recorded in a csv file. The mutual similarity of the recorded values can be seen in the following graph (Figure 11).

#### 3.2.3. Get Train and Get Classification

The Support Vector Machine (SVM) method is a very powerful and versatile machine learning model, capable of performing linear or non-linear classification, regression, and even outlier detection. SVM methods are particularly well suited for classifying complex but small or medium datasets, such as a gesture language dataset.

The fundamental idea behind SVM methods can be better revealed using illustrations. Figure 12 shows part of the gesture dataset 0 and 1. The two classes can be easily and clearly separated with a straight line (they are linearly separable).

The graph on the left shows the decision bounds of the two 1.5 gestures of possible linear classifiers. The model is so poor that it doesn’t even separate classes properly.

For multiple classification problems, we used SVC with an OVO strategy. Although SVM linear classifiers are efficient and work surprisingly well in numerous cases, many datasets are far from being linearly separable. One approach to processing nonlinear datasets involves adding additional features, calculated using a similarity function that measures how much similarity each sample has to a separate landmark.

The technique for solving nonlinear problems involves adding features calculated using the similarity function, which measures how much similarity each sample has to a separate landmark; for example, if we utilize a one-dimensional data set and add a landmark to it at x1 = 0.6, as illustrated by the graph in Figure 13:

Subsequently, we defined the proximity function as a Gaussian Radial Basis Function (RBF) with γ = 0.1 (Equation (1)):(1)$$\phi_{\gamma }(x,\ell )=exp\left(-\gamma ∥x-\ell {∥}^{2}\right)$$

The Gaussian RBF is a bell-shaped function that changes from 0 (very far from the landmark) to 1 (at the landmark). New features can be computed in this way. We use this approach in order to create a landmark based on the location of each sample in the data set. This approach creates many dimensions, and thus increases the chances that the transformed dataset will be linearly separable. If the training set is very large, one will obtain an equally large number of features. SVM allows one to obtain similar results as if multiple proximity features were added, without actually adding them.

We tested (Figure 14) the Gaussian RBF kernel (Gaussian RBF kernel) using the SVC class: svm = SVC (C = 15, gamma = 0.1, kernel = ‘rbf’):

Figure 15 shows models trained with the values of hyperparameters gamma (γ) and C = 41 (dactyl alphabet).

## 4. Results

As part of the paper, a dataset of the alphabet of the Kazakh language, numbers from 1 to 15, with approximately the same depth map, were created. Next, the Get_Data module calculates the coordinates of the joints of the human palm using the BlazePalm palm detector model and creates a data set for one class of gestures. The fastest and most convenient way to store a dataset is the Pickle format, since even a csv file cannot compare with the speed of reading, processing, and viewing pkl files. The SVM is trained using a file, and the model was prepared with a data set of numbers from 1–15 and the alphabets. Two-handed and one-handed gestures were taught separately.

System example: the system was trained based on a data set recorded by a girl of 20 years old, and showed high recognition results at the same distance for children of 8 and 11 years old (Figure 16). In the case where the depth coordinates were changed, the system outputs erroneous classes (Figure 17).

The results obtained during the experiment are clearly shown in the Confusion Matrix table.

The algorithm is usable, but the model does not always perform well because it is filmed in adverse lighting conditions with a weak webcam. Reassembling the model under ideal conditions provides the new possibilities for the program. Therefore, the program is yet to be improved further (Figure 18 and Figure 19).

We use the graph of the confusion matrix to see how the currently selected classifier performs in each class.

Classification accuracy is the accuracy we usually have in mind when we use the term “precision”. We obtain classification accuracy by calculating the ratio of correct predictions to the total number of input samples.

Classification accuracy is good, but gives a false positive sense of achieving high accuracy. The problem arises because the probability of misclassifying samples of a minor class is very high.

In our example, using the dataset of numbers 1–15, and 41 letters of the Kazakh alphabet, the top row shows all data with a true class. The columns show the predicted classes. In the top row, 99% of the numbers 0 of the other numbers are classified correctly; therefore, 99% is the true positive for the correctly classified data in this class, shown in the green cell in the True Positive column. For 99%, we obtained the highest average score of 99.6%, for a dactylic alphabet of 90.6%.

## 5. Discussion

The proposed method recognizes in static and short dynamic gestures in real time, which are represented by two hands, in comparison with the methods described in reference sources considered above.

There are 42 letters in Kazakh Sign Language. Of these, the Kazakh letters i, ң, ғ, k, қ, ө, ë, ъ, and ь are different from the letters of other languages and possess dynamic elements.

In general, the results show the feasibility of the proposed approach to machine learning. In particular, it is shown that the SVM classification model can be trained on the data of a large set of available images, which are processed by the manual control algorithm (MediaPipe Hands) and then successfully tested by the system.

## 6. Conclusions

In this paper, a method based on the palm recognition model and linear recognition models of the dactyl alphabet and sign language numbers is discussed. The method considered was tested experimentally on the data of the magnetic positioning system using a kinematic model of the hand. During the test, the results for letters displayed in two positions were presented. The findings confirm the feasibility of the approach, with approximately 97% classification accuracy.

Therefore, the method enables the development of efficient automated sign language translation systems for sign languages. Such systems are capable of supporting effective human–machine communication and interaction for the deaf and hard of hearing.

Future developments of issues considered in this paper include the application of the method proposed, and experimental setup to solve problems of hand movement recognition.

## 7. Future Work

Firstly, by expanding the system we offer, it is possible to develop a sign language recognition system, which will work on recognizing proper names in the future. Secondly, use of the unified functionality offered in this system can be a prerequisite for developing multimodal videos without the words of sign languages. Thirdly, our system can be a prerequisite for the creation of multimodal sign corpora in the Kazakh language.

## Figures and Tables

**Figure 1 sensors-22-06621-f001:**
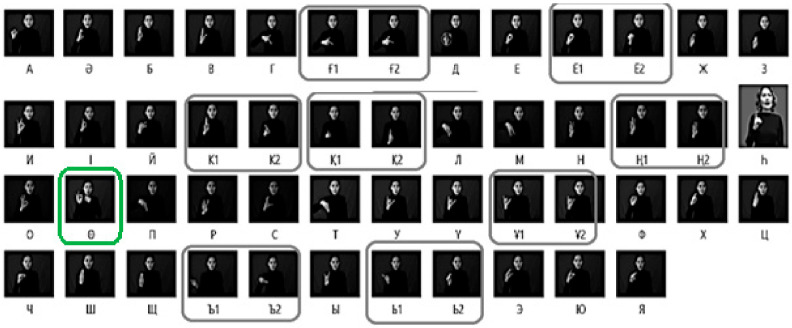
Kazakh Sign Language dactyl alphabet.

**Figure 2 sensors-22-06621-f002:**
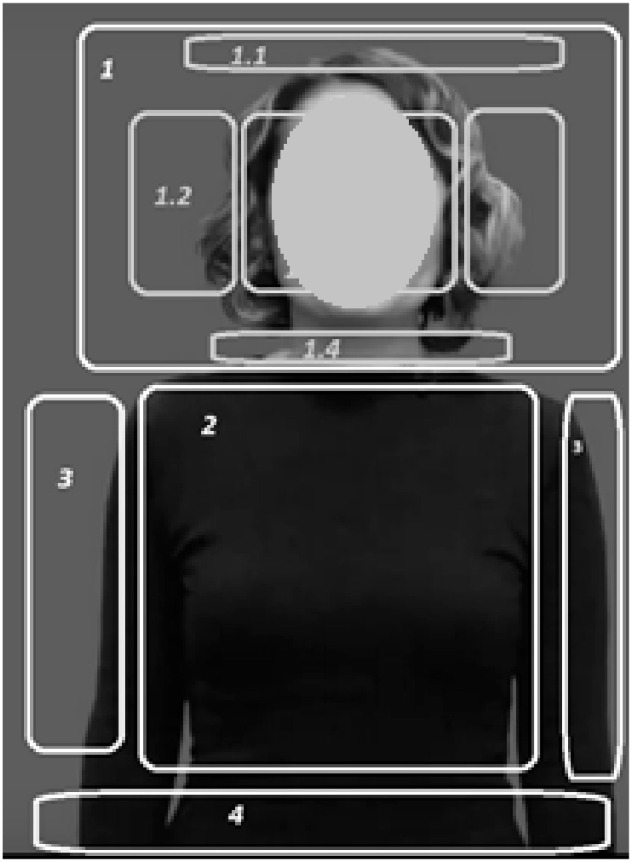
Localization.

**Figure 3 sensors-22-06621-f003:**
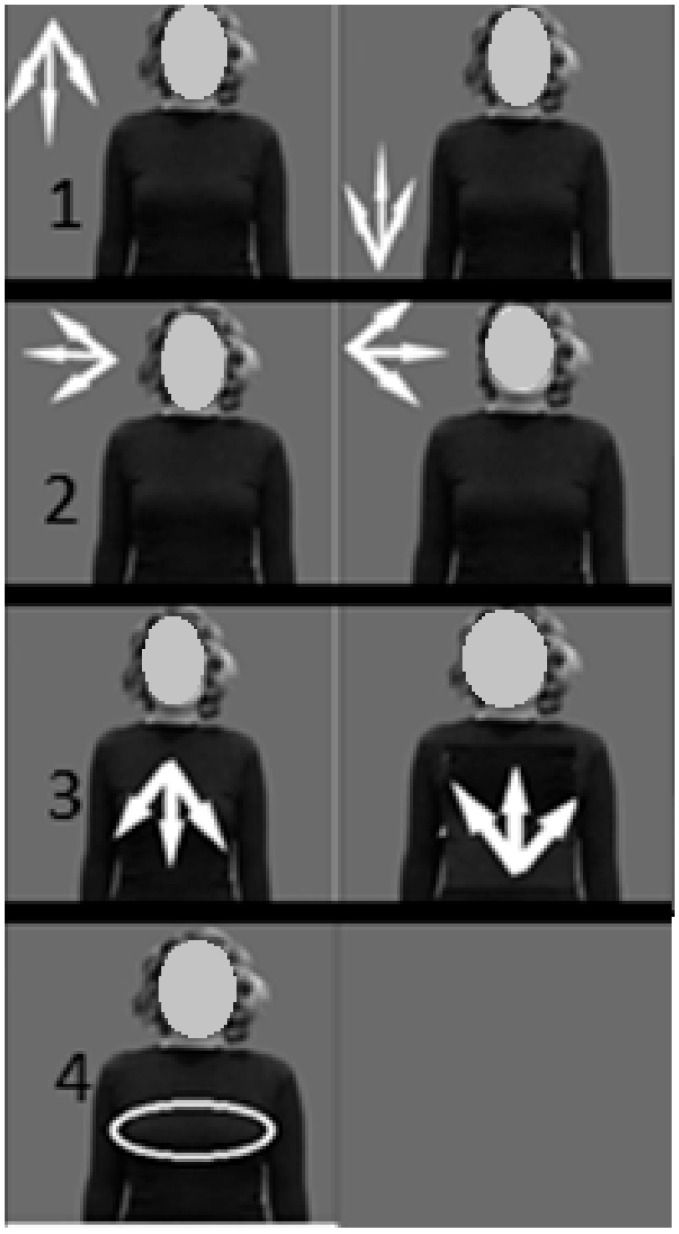
Direction of movement (straight, intermittent, jumping, repetitive).

**Figure 4 sensors-22-06621-f004:**
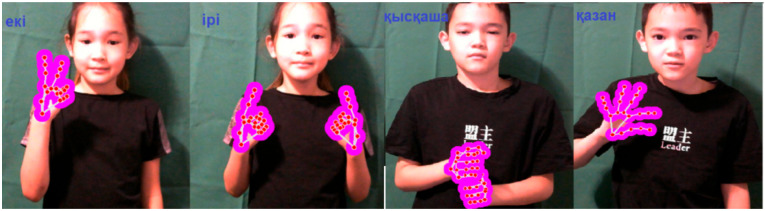
Example of static gestures.

**Figure 5 sensors-22-06621-f005:**
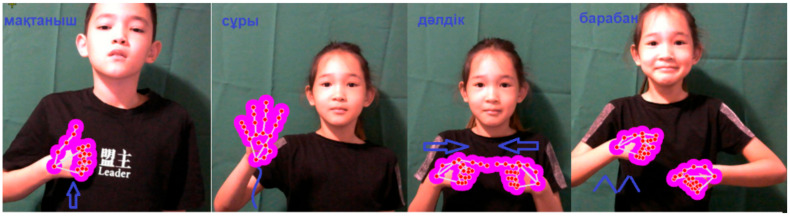
Example of dynamic gestures.

**Figure 6 sensors-22-06621-f006:**
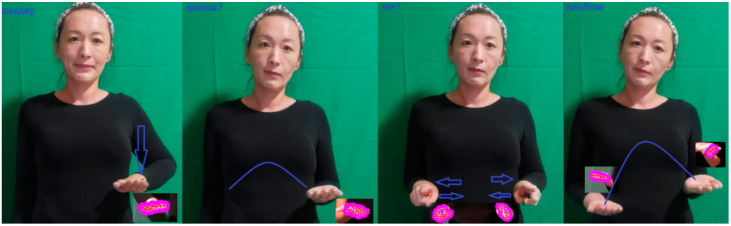
An example of gestures in which it is difficult to recognize the orientation of the palm.

**Figure 7 sensors-22-06621-f007:**
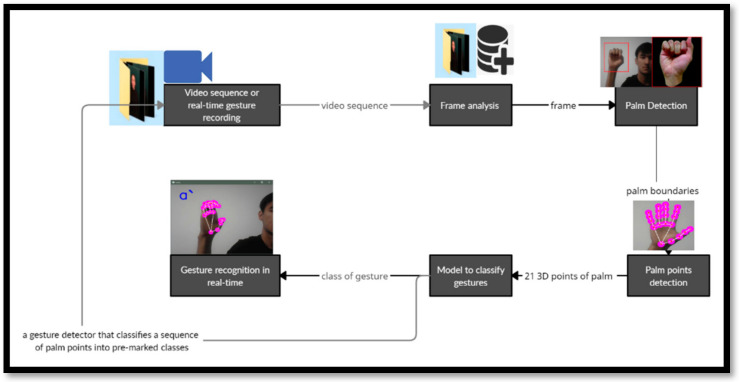
Sign language recognition in a real-time system based on the palm definition model.

**Figure 8 sensors-22-06621-f008:**
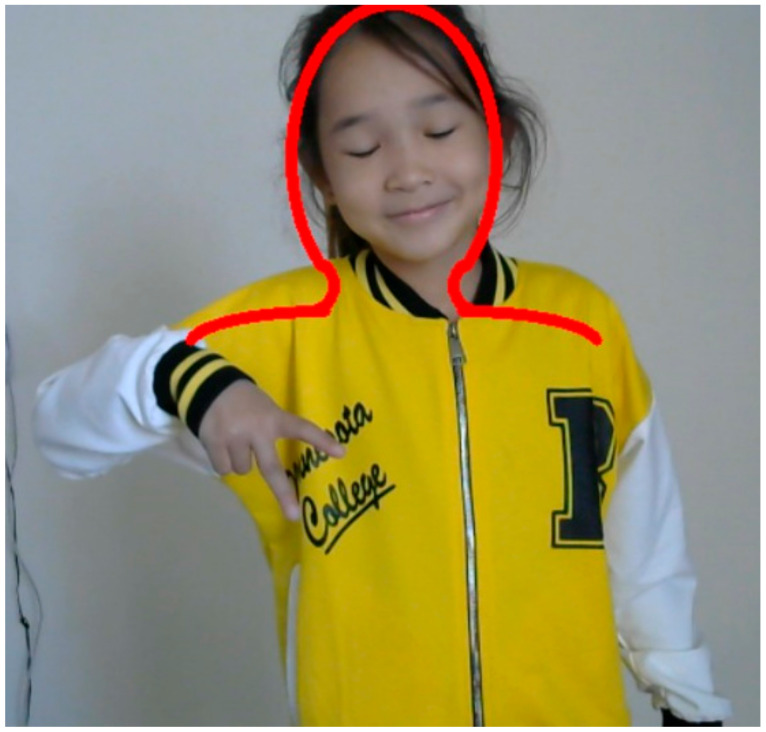
Draw_people functionality.

**Figure 9 sensors-22-06621-f009:**
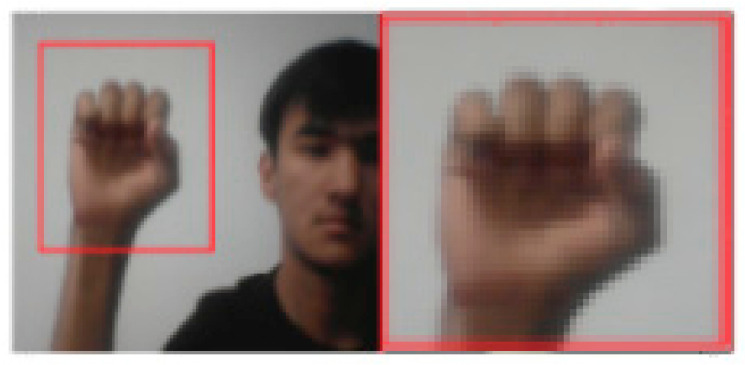
Detected palm.

**Figure 10 sensors-22-06621-f010:**
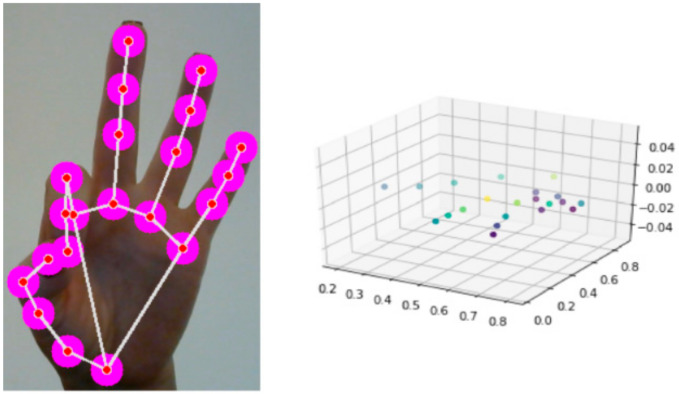
An example of data visualization of some classes.

**Figure 11 sensors-22-06621-f011:**
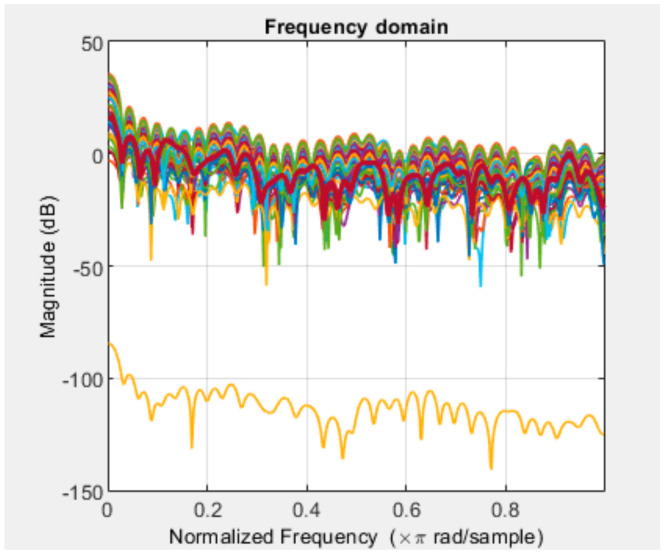
An example of data.

**Figure 12 sensors-22-06621-f012:**
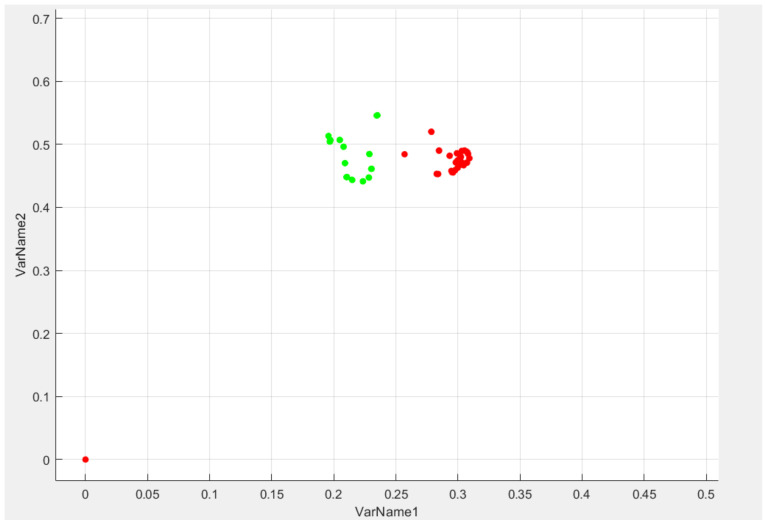
Visualization of gesture data 1.5.

**Figure 13 sensors-22-06621-f013:**
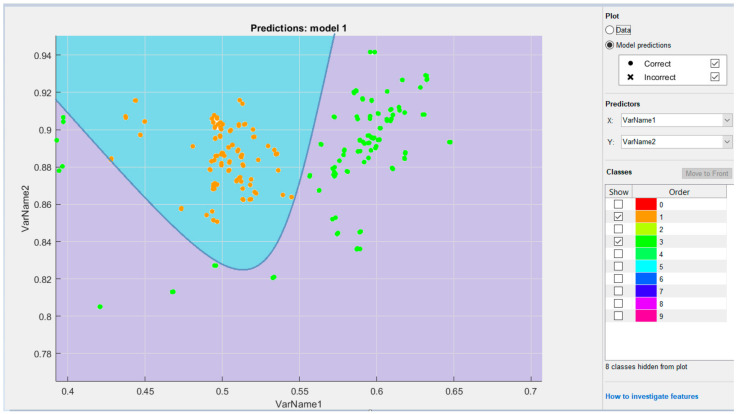
One-dimensional dataset with a landmark.

**Figure 14 sensors-22-06621-f014:**
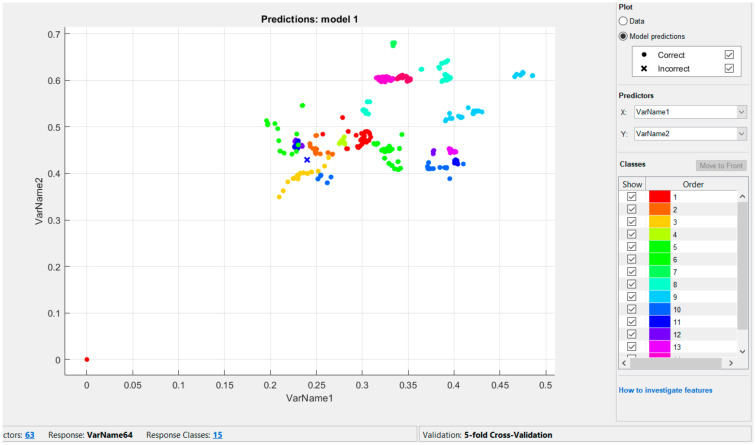
The result of classifying numbers 1–15.

**Figure 15 sensors-22-06621-f015:**
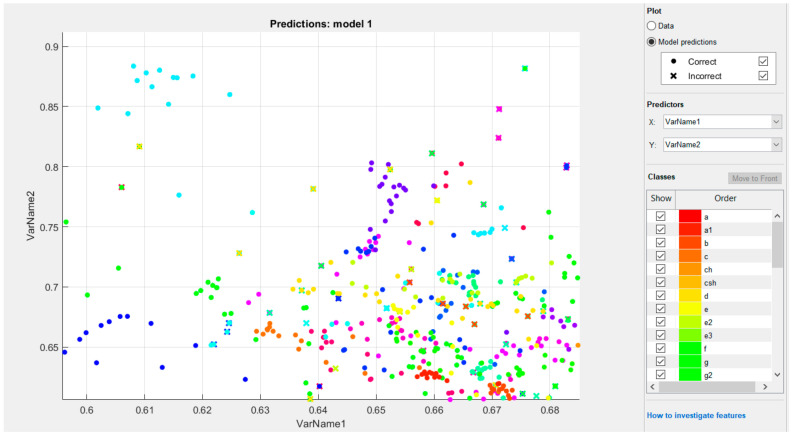
The result of the classification of the Kazakh dactylic alphabet.

**Figure 16 sensors-22-06621-f016:**
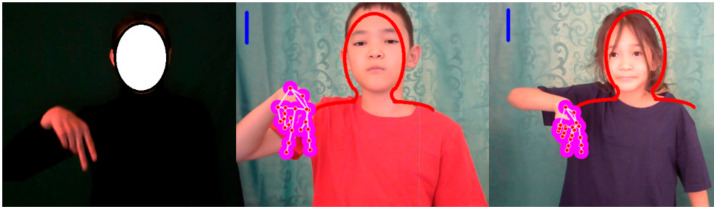
An example of a correctly working system.

**Figure 17 sensors-22-06621-f017:**
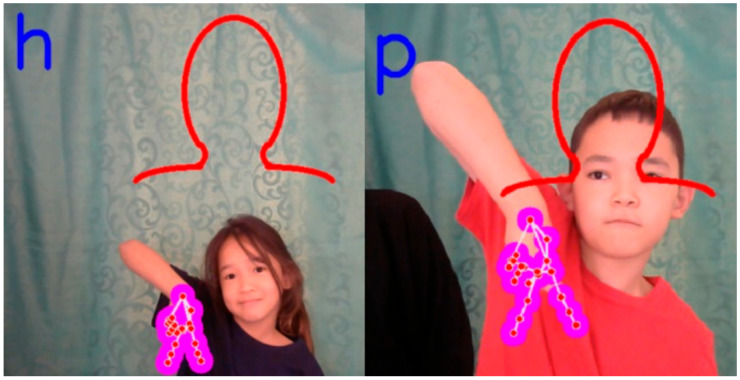
An example of incorrect system operation.

**Figure 18 sensors-22-06621-f018:**
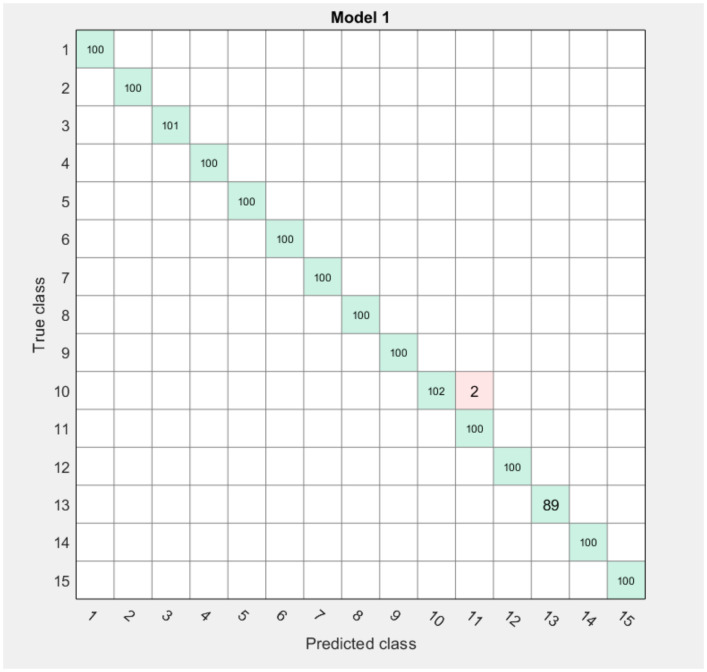
Confusion Matrix of numbers.

**Figure 19 sensors-22-06621-f019:**
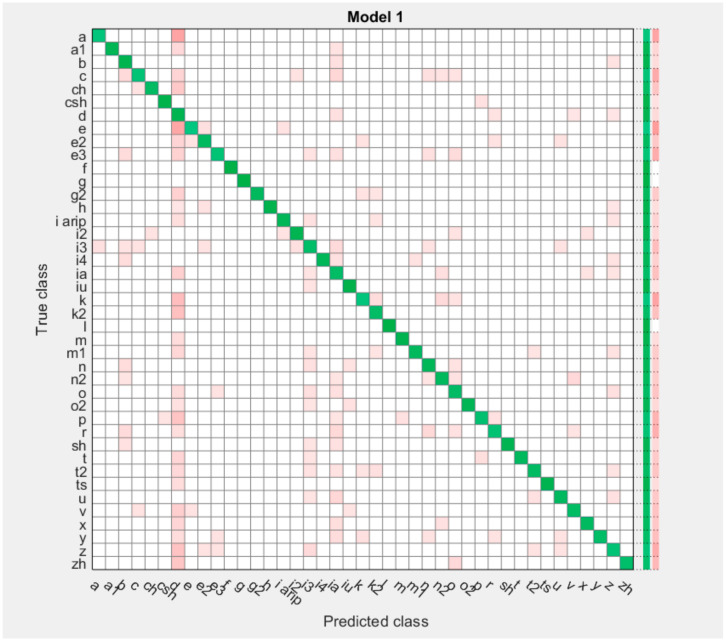
Confusion Matrix of the Kazakh dactylic alphabet.

**Table 1 sensors-22-06621-t001:** The basic parameters of Kazakh Sign Language, received at demonstration.

Localization		Ω
1	in_the_head_area	HA
1.1	over_the_head	HA/OH
1.2	right_left_of_the_head	HA/RLH
1.3	touching_face	HA/TF
1.4	touches_the_neck	HA/TN
2	neutral_zone	NZ
3	near_the_right_shoulder	NRLSH
4	at_the_waist	W
**Palm Orientation**		**ψ**
1	palm_look_right_or_left	PLRLUD
2	palm_looks_to_from_speaker	PTFS
**Direction of Movement**		↔
1	from_to_the_speaker	TFS
2	down_up_ward_Left_right_movement	DULRS
3	circular_movement	CM
4	Motionless	ML
**Types**		**K**
1	one-handed	1H
2	two-handed	2H
2.1	do not intersect-	NTINT
2.2	Intersect	INT

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
