# Peer review of "Sign Language Recognition Method Based on Palm Definition Model and Multiple Classification"

_sensors, 2022, doi:10.3390/s22176621_

Round 1

Reviewer 1 Report

The paper is devoted to many aspects that are related to the technologies of automated recognition of sign languages of deaf people for the implementation of machine sign language translation, as well as the organization of interpersonal and human-machine interaction. The authors of the paper describe in some detail the main problems that arise in the process of solving problems related to automatic machine sign language translation. It can be clearly seen from the paper that despite the great practical potential, the problem of effective recognition of sign languages has not yet been solved due to serious differences in the semantic-syntactic structure of written and sign languages, as a result of which it is not yet possible to perform an unambiguous translation from sign language into, for example text representation. Therefore, there are currently no operating fully automated models and methods for sign language translation systems. To create such full-fledged models, it is necessary to perform a deep semantic analysis and analysis of written phrases, and this is still possible only at a superficial level due to the imperfection of text analysis algorithms and knowledge bases. It is also well noted that the task of improving the quality of life of people with disabilities in the modern information society is given quite a lot of attention. The study of the authors of the paper is aimed at a partial solution of this problem. Testing and comparative analysis of the results of the study were carried out on the collected dataset (numbers and dactyl of the Kazakh sign language). On the basis of comparative analysis, it can be argued that qualitative indicators still depend on the input data. However, in my humble opinion, the paper is not free from a number of shortcomings that need to be addressed. 1) First of all, it is striking that the authors quite extensively describe the problems of machine sign language translation, however, there is little description of the problem that arises, among other things, due to the lack of universal methods for creating sign corpora. Also, the lack of methods and algorithms that improve efficiency machine learning and the accuracy of automatic recognition of sign languages using various video information capture devices that allow you to receive not only high-quality images in the optical mode, but also additional data on the coordinates of graphic areas of interest (depth map mode, infrared mode, etc.). After all, on the basis of such universal methods for creating multimodal gestural corpora, it is possible to create corpora, which in the future can be used, among other things, for the analysis of a non-verbal way of communication (through body movements and hand gestures). And this can lead to the creation of more efficient methods and algorithms that improve the efficiency of machine learning and the accuracy of automatic recognition of sign languages. 2) There are no references to previous works of the world scientific community (2020-22), which are constantly presented at conferences focused on the work of multimodal video analysis (CVPR, ICCV, ECCV, FG, ICASSP, SPECOM and others) or collection of corpora (LREC and others), therefore, it is recommended to expand the section describing previous works, including modern solutions. 3) The analysis of the obtained results (section 4) may need to be supplemented with a detailed description. 4) There are no characteristics of the collected data set, on the basis of which machine learning was performed. 5) In some places of the paper, the authors replace the word "method" with "algorithm", but this is not the same thing. 6) Finally, the style of the paper requires minor revision due to the presence of spelling and punctuation errors.

It seems to me that all the proposed additions will only improve this paper, and it will be useful and interesting to many specialists who associate their research with automatic recognition of not only sign languages, but also human body movements, but only after completion.

In general, the paper should be finalized and then it can be accepted for publication.

Author Response

Thank you so much for the review. Please see the attachment

Reviewer 2 Report

This paper proposes a sign language recognition system based on MediaPipe technology and Support Vector Machine (SVM) model. Specifically, image data are first obtained from a simple camera. Then the coordinates of joints of the human palm are calculated using the BlazePalm single-finger detector model which is available in MediaPipe. Finally, SVM method is used to realize multiple classification of different numbers and letters in Kazakh sign language.

The reviewer has some major concerns that need to be addressed in the next version of this paper as follows.

First, the authors may want to illustrate the novelty of this paper more clearly. This paper mainly adopts MediaPipe technology and SVM method in order to recognize gestures in sign language. However, the basic methodology is quite similar to several other works [40][41][42], which are also based on MediaPipe platform and machine learning models. Furthermore, these works have already achieved relatively high accuracy. Therefore, it would be better, if the authors could point out the essential differences of this paper from other related works and stress its novelty.

Second, the method proposed in this paper can be only applied to certain conditions, in which the place of performance and the palm orientation are restricted. For example, if the user performs gestures in front of the face or neck, then the system becomes invalid because it is difficult to separate the palm from the background. Therefore, the authors may want to explain how to extend the system to more realistic settings.

Third, explanations about the experiments and results are insufficient in this paper. For one thing, this paper lacks basic information about the experimental settings including experimental scenarios, experimental devices, sample size, and so on. Detailed information about the experiments could help increase the validity of the experiments and the reliability of the results. For another, the authors may want to further provide some discussions about the results. For instance, it can be seen from Figure 16 that the accuracy of classifying certain classes, i.e. numbers 2 and 6, is low. It would be better if the authors could provide some possible reasons for this.

Fourth, the authors fail to cite properly several past literatures (e.g., [1-3]) highly related to this work, and clearly discuss the differences between them and this paper.

[1] SignSpeaker: A Real-time, High-Precision SmartWatch-based Sign Language Translator, MobiCom 2019.

[2] SKEPRID: Pose and Illumination Change-Resistant Skeleton-Based Person Re-Identification, TOMM 2020.

[3] Finger Gesture Tracking for Interactive Applications: A Pilot Study

with Sign Languages, IMWUT 2020.

Fifth, it would be better, if the authors could refine the presentation of this paper. For example, Figure 12, Figure 13, Figure 16 could be enlarged for better visualization. The structure of the workflow in Figure 7 could be slightly revised to make it clearer.

Author Response

Thank you so much for the review. Please see the attachment.

Round 2

Reviewer 2 Report

The authors have addressed my comments to the previous version. I do not have further comments and recommend that this paper be accepted.